# Spatiotemporal Dynamic of COVID-19 Diffusion in China: A Dynamic Spatial Autoregressive Model Analysis

Hanchen Yu [1], Jingwei Li [2,3,*], Sarah Bardin [4], Hengyu Gu [5] and Chenjing Fan [6]

1   Center for Geographic Analysis, Harvard University, Cambridge, MA 02138, USA; hanchenyu@fas.harvard.edu
2   School of Architecture and Design, Beijing Jiaotong University, Beijing 100044, China
3   School of Architecture, Tsinghua University, Beijing 100084, China
4   Spatial Analysis Research Center, School of Geographical Sciences and Urban Planning, Arizona State University, Tempe, AZ 85281, USA; sfbardin@asu.edu
5   School of Government, Peking University, Beijing 100871, China; henry.gu@pku.edu.cn
6   College of Landscape Architecture, Nanjing Forestry University, Nanjing 210037, China; fancj@njfu.edu.cn
*   Correspondence: lijingwe17@mails.tsinghua.edu.cn

**Abstract:** COVID-19 has seriously threatened people's health and well-being across the globe since it was first reported in Wuhan, China in late 2019. This study investigates the mechanism of COVID-19 transmission in different periods within and between cities in China to better understand the nature of the outbreak. We use Moran's I, a measure of spatial autocorrelation, to examine the spatial dependency of COVID-19 and a dynamic spatial autoregressive model to explore the transmission mechanism. We find that the spatial dependency of COVID-19 decreased over time and that the transmission of the disease could be divided into three distinct stages: an eruption stage, a stabilization stage, and a declination stage. The infection rate between cities was close to one-third of the infection rate within cities at the eruption stage, while it reduced to zero at the declination stage. We also find that the infection rates within cities at the eruption stage and declination stage were similar. China's policies for controlling the spread of the epidemic, specifically with respect to limiting inter-city mobility and implementing intra-city travel restrictions (social isolation), were most effective in reducing the viral transmission of COVID-19. The findings from this study indicate that the elimination of inter-city mobility had the largest impact on controlling disease transmission.

**Keywords:** COVID-19; spatial dependency; dynamic spatial autoregressive model; spatial diffusion

## 1. Introduction

In December 2019, coronavirus disease 2019 (COVID-19) was first reported in Wuhan, China. Soon after its discovery, COVID-19 began to spread rapidly to other provinces across China, reaching its peak in early February 2020. Despite its severity and subsequent designations as a Public Health Emergency of International Concern (PHEIC) and a global crisis [1], the epidemic in China was generally well-controlled by the end of February. In fact, on 18 March 2020, China marked its first day of no new confirmed cases [2]. Despite occasional small-scale concentrated outbreaks and the overseas import of COVID-19 throughout 2020, widespread outbreaks of the disease never reoccurred. As of the writing of this paper, more than 27.5 million cases of COVID-19 have been confirmed worldwide, and the death toll is approaching 2.4 million [1]. Given the unprecedented impact of this pandemic, it is worthwhile to explore both the spatial–temporal variation of the COVID-19 epidemic within China and the underlying mechanisms which drove the disease to spread throughout the country.

Previous studies have shown that the prevalence of COVID-19 is influenced by many factors, including environmental, socioeconomic, demographic, and migratory patterns [3–5]. For instance, it has been hypothesized that temperature and humidity

may contribute to the spread of COVID-19 [6] due to moderate, diurnal temperatures and low humidity creating favorable conditions for viral diffusion [7,8]. Additionally, research shows that long-term exposure to air pollution may intensify the spread of the virus, as aerosol droplets emitted by persons with COVID-19 during sneezing, coughing, or simply talking are stabilized in the air through coalescence with particulate matter (PM) at high concentrations and under conditions of atmospheric stability [3,9,10]. The transmission of COVID-19 may also be influenced by individual characteristics, such as age and gender, population size and density, family composition, occupation, and income, among other socioeconomic and demographic factors [11–15]. Moreover, migratory and population flows through urban centers may also affect the speed and pattern of virus transmission [16–19]. In China, for example, the frequency of airline and high-speed railway service outside of Wuhan is significantly correlated with the number of COVID-19 cases in the destination city; that is, the further away from Wuhan, the lower the case number and transmission velocity [20].

Despite the wealth of COVID-19 publications in the past year, much of this research to date has relied on correlation- and regression-based techniques, focusing on simple linear relationships between COVID-19 transmission and risk factors [21–23] with limited studies accounting for the underlying mechanism that drives the dynamic spatial diffusion of COVID-19 [10,22,24,25]. While a combination of anthropogenic causes, including socioeconomic and demographic factors and natural factors, such as temperature, humidity, and atmospheric conditions, may affect the transmission rate, these factors alone do not adequately account for the diffusion of the virus across space. Previous studies emphasized the important role of transportation factors in the spread of COVID-19 [19]. However, those were unable to accurately quantify the spillover effects between cities. On the other hand, many studies discuss the epidemic transmission pattern and developing trend based on the SIR model from the perspective of epidemic dynamics, which focuses more on its spatial–temporal evolution rather than the effect of different policies on virus transmission [4,26,27].

Research suggests that China's ability to control the epidemic effectively was mainly the result of improved medical services, prevention and control measures within the community, and the restriction of population movement between cities [28–30]. Virus transmission in China has shown a clear pattern of 'erupt-stabilize-decline'. In fact, each phase reflects China's policies at different times of the epidemic. Previous studies, however, lacked quantitative analysis on the COVID-19 transmission mechanism in different periods within and between cities. Our goal is to study the spread of the epidemic between cities by adding spatial spillover effects. This quantitative study helps to understand the spread of the epidemic between cities. It may contribute to epidemic prevention and control and improve public health governance.

Using data from 280 cities in China, this paper studies new cases of COVID-19 in different transmission stages and reveals the spatial and temporal variation in the number of new cases at the city scale. First, we analyze the temporal and spatial clustering characteristics of new cases in different cities based on the spatial autocorrelation model, which informed the classification of the transmission process into three stages: an epidemic eruption stage, an epidemic stabilization stage, and an epidemic declination stage. A linear regression model is then used to analyze the influencing factors of daily new cases and explore the impact of Wuhan population migration on the epidemic transmission. Finally, we apply a dynamic spatial autoregressive (DSAR) model based on the susceptible-infected-removed (SIR) model to analyze the spatial and temporal components of the number of new cases. We discuss the correlation between new cases in surrounding and local cities and the epidemic situation in local cities. We conclude with relevant policy recommendations based on China's anti-epidemic experience.

## 2. Research Data and Methods

### 2.1. Exploratory Spatiotemporal Analysis

Data on the number of daily COVID-19 cases for each city were obtained from the National Health Commission and the Provincial Health Commissions. These data include information from 25 January 2020 to 13 March 2020. We analyzed the spatiotemporal pattern of the daily and average new cases. To more rigorously evaluate the spatial autocorrelation of the daily news cases and their change over time, we performed a Moran's I test. Moran's I is a classic autocorrelation measurement, which can be modeled as follows. Consider a variable x distributed over a set of $N$ locations, each labeled *i*. Moran's I is then defined as

$$I = \frac{N \sum_{i=1}^{N} \sum_{j=1}^{N} w_{ij}(x_i - \overline{x})(x_j - \overline{x})}{\sum_{i=1}^{N} \sum_{j=1}^{N} w_{ij} \sum_{i=1}^{N}(x_i - \overline{x})} \tag{1}$$

where $N$ is the number of locations indexed by *i* and *j*; $\overline{x}$ is the mean of x; and $w_{ij}$ is a matrix of spatial weights. We described the time series of new cases and the corresponding Moran's I time series. In this way, we found the stage of development of the epidemic and the temporal trend of its spatial pattern. We used the Matlab spatial econometrics toolbox to calculate Moran's I for each period cross-section [31].

### 2.2. Classical Linear Model

To study what factors may have affected the spread of the epidemic, we regressed the average daily new cases on a variety of environmental, demographic, socioeconomic, and migratory factors (Table 1) using ordinary least squares (OLS). Table 1 also shows the data sources of these factors. We used Stata to run the OLS estimation of the latter two aspatial models. To compare the impact of migration from Wuhan, we used two models:

(1)     We regressed on all cities but excluded the variables 'In' and 'Out'.
(2)     We regressed on cities other than Wuhan and included variables 'In' and 'Out', as those two variables were only for cities other than Wuhan.

**Table 1.** Explanatory variables used in this study together with sources.

| Category | Variable Name | Description | Source [32] |
|---|---|---|---|
| **Environmental** | AQI | Average air quality index during the epidemic | Ministry of Ecology and Environment of People's Republic of China |
| - | Humidity | Average humidity during the epidemic (%) | China Meteorological Administration |
| - | Temperature | Average temperature during the epidemic (celsius) | China Meteorological Administration |
| - | PM2.5 | Annual mean concentration of PM 2.5 ($\mu g/m^3$) | China city statistical yearbook 2019 |
| **Demographic** | Population | Annual average population (millions of people) | China city statistical yearbook 2019 |
| - | Density | Population density of each city (100 people/km$^2$) | China city statistical yearbook 2019 |
| - | Household | Average people per household (people per household) | China city statistical yearbook 2019 |
| - | Age | Average age of residents (years) | Sixth National Population Census of China |
| - | Ethnic | Proportion of racial and ethnic minorities (%) | Sixth National Population Census of China |
| - | Gender | Percentage of males (%) | Sixth National Population Census of China |
| - | Education | Average years of education (years) | Sixth National Population Census of China |
| - | Urban | Non-agricultural household population proportion (%) | Sixth National Population Census of China |
| **Socioeconomic** | Wage | Average annual wage of employed staff and workers (10,000 yuan) | China city statistical yearbook 2019 |
| - | Insurance | Percentage of employees joining urban basic medical care system (%) | China city statistical yearbook 2019 |
| - | Unemployment | Percentage of unemployment (%) | China city statistical yearbook 2019 |
| - | GDP | Gross domestic product (100 million yuan) | China city statistical yearbook 2019 |
| **Migration** | Bus | Annal highway passenger traffic (100 million people) | China city statistical yearbook 2019 |
| - | Ship | Annal waterway passenger traffic (100 million people) | China city statistical yearbook 2019 |
| - | Air | Annal civil aviation passenger traffic (100 million people) | China city statistical yearbook 2019 |
| - | In | Probability of the migrant inflow between Wuhan and each city (%) [19] | China Migrant Population monitoring data 2017 |
| - | Out | Probability of the migrant outflow between Wuhan and each city (%) | China Migrant Population monitoring data 2017 |

### 2.3. Dynamic Spatial Autoregressive Model

However, OLS is a cross-sectional model and thus could not study the temporal and spatial dynamics of the number of new cases. Moreover, the OLS model is not an

epidemiological model, meaning it does not explain the mechanism of infectious diseases. Epidemiological models of the SIR type describe disease spread dynamics based on three main factors: the size of the population, the number of susceptible individuals, and the number of infected individuals [33]. With a population of size N, if I denotes the number of infected individuals and R denotes the number of recovered individuals, then the number of individuals susceptible to the disease is given by S = N − I − R. At each time t, the number of new infections will depend on the interactions of the susceptible (S) and infected (I) individuals. Infected individuals are typically non-infectious during the latent period of their disease and asymptomatic, but infectious from the end of the latent period to the end of the incubation period and then become infectious with symptoms after the incubation period. If j denotes the number of days it takes to become infectious, at time t, the interactions of susceptible people with people infected $t − j$ days earlier may lead to new cases.

Using daily reports of coronavirus cases for cities across China, we generated a panel dataset of Chinese cities over 49 days from 25 January 2020 to 13 March 2020. With this panel data, we were able to control for city-specific fixed effects in our estimation of disease transmission. The panel estimated the number of cases as a function of the potential pool of susceptible and infected individuals and time and city-specific effects, and this is given by the following equation:

$$y_{it} = \beta_1 y_{it-1} + \beta_2 \frac{S_{it} * I_{i,t-j}}{N} + \gamma_i + \delta_t + \varepsilon_{it} \tag{2}$$

where $y_{it}$ denotes the number of reported cases in city $i$ at time $t$, $\gamma_i$ gives the fixed effect parameter for city $i$, $\delta_t$ is the fixed effect parameter for time $t$, and $\varepsilon_{it}$ is the error for city $i$ at time $t$.

In comparison with $N$, $I$ is small. The $\frac{S_{it}*I_{i,t-j}}{N}$ term is near zero, and Equation (2) can be expressed as a typical dynamic panel model:

$$y_{it} = \beta_1 y_{it-1} + \gamma_i + \delta_t + \varepsilon_{it} \tag{3}$$

However, the typical dynamic panel model does not contain information related to the spread between cities. This means that the spread of COVID-19 between cities can only be measured with the help of traffic factors. It is hard to know the number of people involved in inter-city traffic who had COVID-19. A common assumption is that the proportion of COVID-19 in the population was the same as the origin. This obviously overestimates the proportion of infections among people in inter-city traffic, since most patients had difficulty traveling to other cities during their illness. On the other hand, traffic data usually have large errors. All this makes it difficult to estimate the spread between cities. The DSAR model is an extension of the classic SIR model in space. It only uses the case number in each city to directly reflect the transmission between cities, and the case number is more accurate than traffic data. Therefore, the DSAR model is more powerful in describing the spread of the epidemic between cities. The DSAR model we employed is expressed as follows:

$$y_{it} = \beta_1 y_{it-1} + \rho_1 W y_{it} + \rho_2 W y_{it-1} + Xb + \gamma_i + \delta_t + \varepsilon_{it} \tag{4}$$

where W is a weights matrix and $X$ is the control variables. The Weights matrix $W = \{w_{ij}\}$ is the migration ratio between cities during the epidemic period. We used row normalization on the matrix to make the sum of each row equal to 1. The migration data were from Baidu Mobility Data. Hence, $W y_{it}$ represents the weighted average number of reported cases for cities that have migrated to city $i$ at time $t$. Since the demographic, socioeconomic, and migration variables were cross-sectional, we used daily environmental variables as the control variables. We used the Matlab spatial econometrics toolbox and Jihai's DSAR code [34] to run the maximum likelihood estimation of the DSAR model.

## 3. Results

### 3.1. Spatiotemporal Pattern of New Cases

The spatial distribution of the average daily new cases is shown in Figure 1. Figure 1 shows that the average new cases were mainly distributed in cities in Hubei Province. There was pronounced spatial autocorrelation with respect to the average number of new cases, as evidenced by the strong clustering of similar colors across space. Therefore, it can be inferred that the virus exhibited significant spatial diffusion.

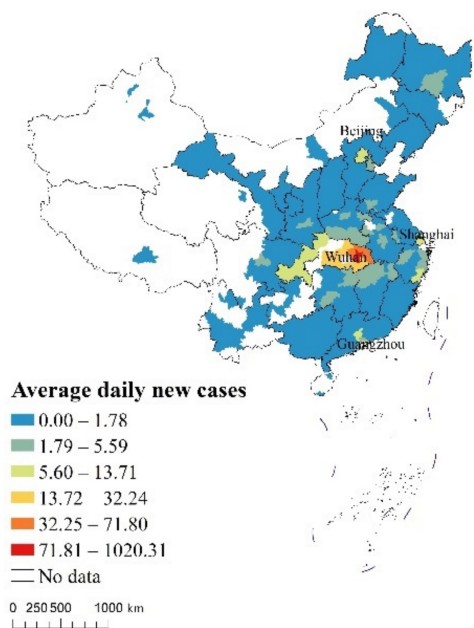

**Figure 1.** The maps of average daily new COVID-19 cases.

The spatial distribution of daily new cases is shown in Figure 2. Figure 2 illustrates the changes in new cases over time. The number of new cases increased from day 1 to day 10 and plateaued from day 11 to day 18 before gradually decreasing from day 19 to day 49. The distribution of new cases across the country on the first day was relatively scattered. However, as time went on, the distribution of new cases became increasingly concentrated in Hubei Province, especially within Wuhan City. The distribution of the average new cases was similar to the distribution of the new cases on Day 10.

The line graph of the daily new cases with Moran's I is shown in Figure 3. The jump on the day 19 was due to the revision of the diagnostic criteria (including clinical cases). Given this temporal pattern, we divided the pandemic into three distinct stages: the eruption stage (day 1–10), the stabilization stage (day 11–18), and the declination stage (day 19–49). The dynamic panel model we used was also divided into those three time periods. Figure 3 also shows the Moran's I for daily new cases. In general, the Moran's I decreased over time, indicating that the spread of the epidemic between cities decreased. Although the epidemic had an obvious urban-scale spatial autocorrelation at the beginning, this spatial autocorrelation gradually decreased over time after day 10. This shows that the spread of the epidemic between cities was under control after day 10. Considering that the lockdown policy went into effect on 23 January, it took about two weeks from the beginning of the policy to stabilize the epidemic.

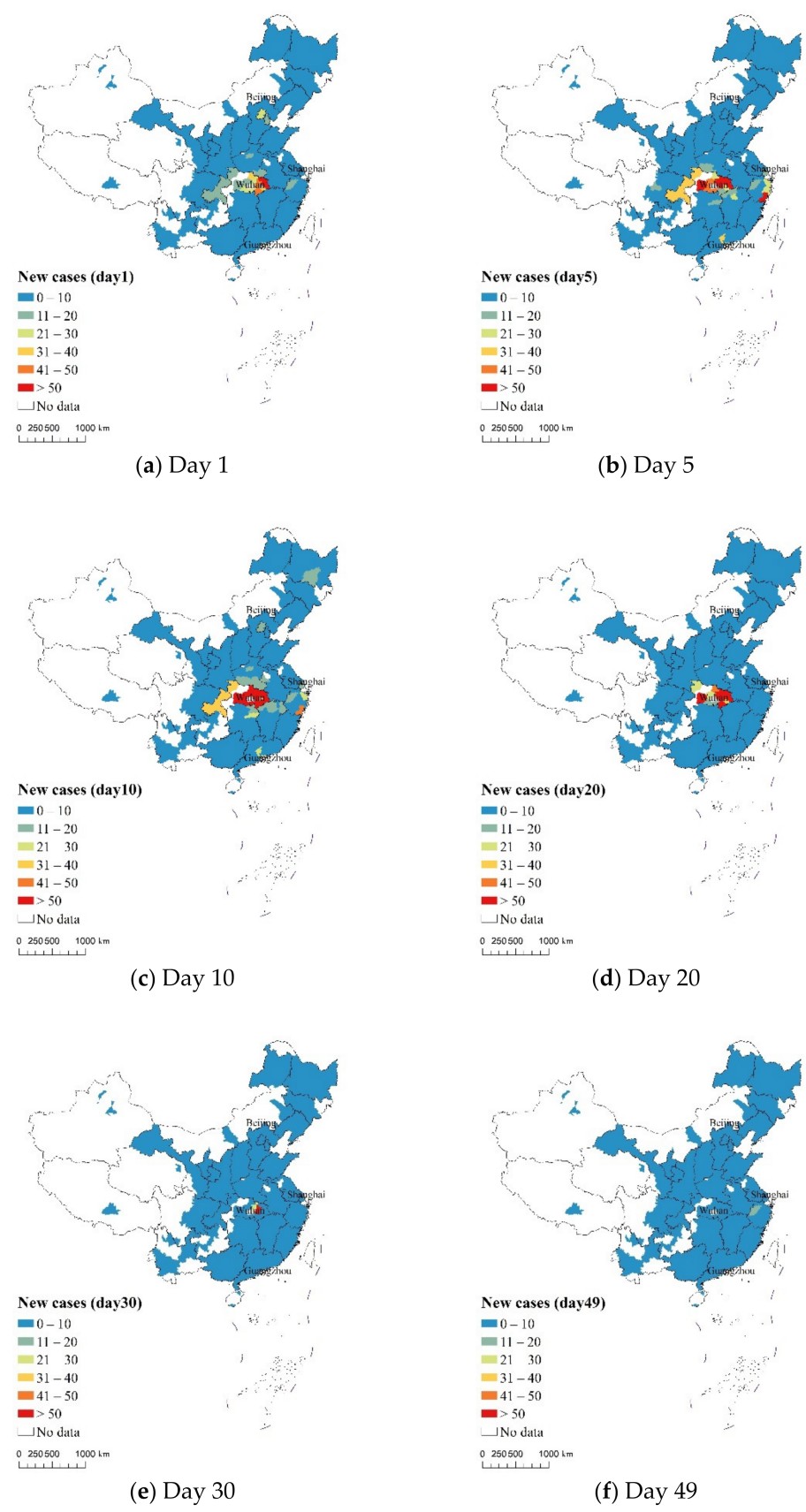

**Figure 2.** The maps of daily new COVID-19 cases. (**a**–**f**) are the daily new cases distribution from Day 1 (25 January) to Day 49 (13 March).

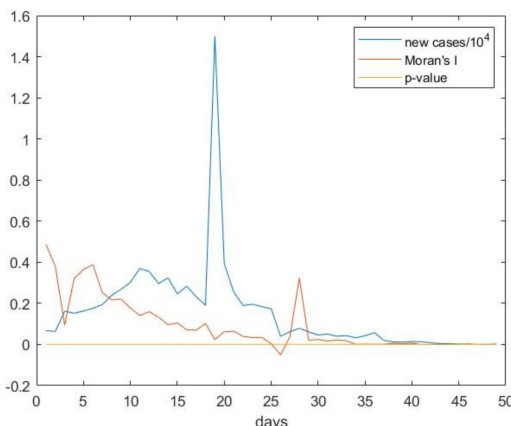

**Figure 3.** The daily new cases and Moran's I.

### 3.2. The Impact of Migration on Virus Transmission

The OLS parameter estimates are shown in Table 2. The R2 of the regression results for all cities was 0.07, which means the model had low explanatory power, while the R2 of cities other than Wuhan was 0.9, which means the goodness of fit was high. Since Model 2 included the variables 'In' and 'Out', the explanatory power of the model was greatly improved, indicating that migration from Wuhan was a driving factor in determining the number of cases. For every 1% increase in the probability of commuting to Wuhan, there was an average expected increase of 3.66 cases per day. Meanwhile, for every 1% increase in the probability of commuting out of Wuhan, there was an average expected increase of 1.13 cases per day. Among all the other variables, only the GDP was significant. For every 100 million yuan increase in GDP, the number of new cases increased by 3.43. In summary, the higher the GDP, and the stronger the migratory relationship with respect to Wuhan, and the higher the expected average number of cases. All other factors had no statistically significant impact on the spread of COVID-19. The comparison of OLS shows that the connection with Wuhan was the main factor in determining the number of new cases.

**Table 2.** Parameter estimates of OLS.

| Explanatory Variable | Model 1: All Cities | Model 2: Cities Other Than Wuhan |
|:---:|:---:|:---:|
| | R2 = 0.07 | R2 = 0.90 |
| | | Coefficient |
| AQI | −0.52 | −0.01 |
| Humidity | −0.22 | 0.01 |
| Temperature | 0.35 | 0.03 |
| PM2.5 | 0.37 | 0.03 |
| Population | 0.59 | −0.08 |
| Density | −0.20 | −0.10 |
| Household | 4.89 | 0.36 |
| Age | 6.42 | −0.29 |
| Ethnic | −0.25 | 0.04 |
| Gender | −0.14 | 0.00 |
| Education | 0.15 | 0.02 |
| Urban | 0.52 | 0.02 |
| Wage | −3.45 | −0.29 |
| Insurance | −3.88 | −0.28 |
| Unemployment | 0.23 | 0.04 |
| GDP | 45.80 * | 3.43 ** |
| Bus | −0.11 | 0.04 |
| Ship | −0.72 | 0.03 |
| Air | −1.11 | −0.01 |
| In | - | 3.66 ** |
| Out | - | 1.13 ** |
| Intercept | −26.44 | −0.81 |

Notes: * $p < 0.05$; ** $p < 0.01$.

### 3.3. Dynamic Spatial Diffusion of the COVID-19 Epidemic

In order to describe the impact of Wuhan migration on the spread of the epidemic in more detail, we employed a DSAR model. The analysis results are shown in Table 3. The term y(t−1) indicates the impact of the city's previous day's new additions on the focal day, W*y(t−1) refers to the impact of the previous day's surrounding areas on the focal day, and W*y refers to the impact of new cases in surrounding cities on the focal day. Environmental factors had no effect on the spread of the epidemic. The spread of the epidemic depended only on the number of new cases in the local and surrounding areas. During the eruption stage (days 1–10), the infection coefficient within the city was less than 1; specifically, for every positive case identified within the city during the previous day, 0.76 new cases were expected. In contrast, for every new case in surrounding cities the day before, the local increase was 0.16, the surrounding cities increased by 1 on the following day, and the local cases increased by 0.08. By summing these two effects, we found that the infection coefficient between cities was 0.24, and the total infection coefficient within a city and between cities was 1. During the outbreak period, although the disease within the city was controlled, the number of new cases increased due to the disease occurring between cities. In the epidemic stabilization stage, the infection between cities remained the driving factor. The infection within the city was not significant, and the R2 of the model was low, which may have been caused by the different statistical methods of different cities at this stage. In the epidemic declination stage, the infection coefficient inside the city was 0.74, which was basically the same as the epidemic eruption stage. However, the spread between cities was under control. For every case increase of 1 in the surrounding area the day before, the local cases decreased by 0.05, and for every case increase of 1 in the surrounding area that day, the local new cases increased by 0.05. The spillover effects between cities canceled each other out. The total infection coefficient was less than one, leading to a gradual decrease in the number of new cases. Overall, the reason why the epidemic in China was controlled was due to the interruption of the spread between cities, causing the total infection coefficient to be less than one. During the epidemic, the infection coefficient within a city was basically the same.

**Table 3.** Parameter estimates of the DSAR model.

| Explanatory Variable | Day 1–10 (R2 = 0.26) | Day 11–18 (R2 = 0.07) | Day 19–49 (R2 = 0.85) |
|:---:|:---:|:---:|:---:|
| | Coefficient | Coefficient | Coefficient |
| y(t−1) | 0.76 ** | −0.04 | 0.74 ** |
| W*y(t−1) | 0.16 ** | 0.46 ** | −0.05 ** |
| W*y | 0.08 * | 0.28 ** | 0.05 ** |
| AQI | 0.00 | 0.00 | 0.00 |
| Humidity | 0.00 | −0.03 | −0.02 |
| Temperature | −0.06 | −0.20 | 0.01 |

Notes: * $p < 0.05$; ** $p < 0.01$.

Finally, the three robustness tests of the model were all significant, as shown in Table 4 below. The Levin–Lin–Chu unit root test for y showed that the dependent variable did not have a unit root and was a stable process. The Kao test clearly showed that the dynamic panel model was cointegrated. The Chow test clearly showed that the division of the epidemic into these three stages was reasonable.

**Table 4.** Robustness testing.

| Test | Statistics | *p*-Value |
|:---:|:---:|:---:|
| Levin–Lin–Chu unit root test for y | −23.3338 | 0 |
| Kao test for cointegration | −74.5425 | 0 |
| Chow test for breaks | 10.0937 | 0 |

## 4. Discussion

### 4.1. Spatial and Temporal Characteristics of the Epidemic Situation in China

This paper examined newly confirmed cases of COVID-19 in China at the city scale, particularly with respect to spatial and temporal dimensions. Temporally speaking, the total number of new cases in China from 25 January to 13 March presented a gradually decreasing trend and generally experienced three stages. With the emergence of the first confirmed cases in mid-January 2020, Chinese authorities put Wuhan on lockdown, suspending all means of transport in and out of the city beginning on 23 January. Daily new confirmed cases continued to increase and peaked on 4 February, making the period from 25 January to 3 February the first phase of the epidemic, which we refer to as the eruption stage (day 1–10). Between day 11 and day 18, the number of new daily confirmed cases showed a fluctuating downward trend, marking the stabilization stage (4 February–11 February), which was achieved under the effective implementation of national epidemic prevention and control measures (for instance, quarantining the infected persons and tracing their close contacts). The third phase began with a sharp increase on 12 February due to the revision of the diagnostic criteria (including clinical cases), but following strengthened prevention and control measures, a gradual decrease in confirmed cases was achieved. There were fewer than 300 cases on day 30 (23 February) and only 23 new cases by day 49 (13 March), marking the third and final declination stage (day 19–49). It can be seen that the epidemic curve in China reached a peak on 4 February (12 days after the Wuhan lockdown), declined rapidly after the peak value, and was brought under control in only about 50 days. Based on available public data, the first instance of zero growth of COVID-19 cases appeared on 18 March 2020. Although there was small growth after that, such as in Beijing in June 2020 and Guangdong Province in May 2021, Most of the cases were imported cases from abroad, and the epidemic was controlled in the city where the outbreak occurred. COVID-19 did not spread between cities after 18 March 2020. As our goal was to study the spread of the epidemic between cities, we only focused on the first wave of COVID-19 in China.

From the spatial dimension, the distribution of new cases in China primarily presented a Wuhan-centered clustering feature that became more and more concentrated in a few cities over time. The epidemic spatial correlation at the city scale appeared to be lower and lower, and the spatial transmission in general could also be categorized into three phases: aggregation, diffusion, and degradation. The outbreak quickly spread to almost all provinces and cities in eastern and central China on day 1 (25 January). During the aggregation phase, the greatest number of new cases centered on cities in Hubei Province and megacities like Beijing, with Wuhan as the centrum of new cases. These cities, benefiting from convenient access and especially geographical proximity, all share muscular intercity mobility and close connections with Hubei Province in terms of traffic patterns. During the diffusion phase from day 5 (30 January) to day 10 (4 February), new cases showed a decreasing gradient from Wuhan and Hubei Province to the surrounding area, making a region-centered epidemic spatial pattern (a severe outbreak was concentrated more in regions such as Hubei Province, the Yangtze River Delta, Pearl River Delta, Beijing-Tianjin-Hebei, and Chengdu-Chongqing). In addition, the significant reduction in the number of cities identified as the center of new outbreaks indicated the gradual control of transmission between cities. The epidemic transmission entered the degradation phase after day 20 (13 February), with only a few new cases identified in Hubei Province, which possibly revealed an intra-city transmission pattern rather than inter-city transmission between Hubei and the surrounding provinces. On day 30 (22 February), new cases emerged in cities outside of Hubei Province (such as Qingdao, Tonghua, and Nanchong), indicating significantly decreased spatial clustering of the epidemic. The number of infected cities also remained stable. The inter-city transmission was by and large controlled by day 49 (13 March), when new cases only remained in Ezhou, Hangzhou, Shanghai, and Hebei Province.

### 4.2. Factors Influencing Epidemic Transmission

In order to explore the causes of COVID-19 transmission, we used an OLS regression model. The findings from this analysis showed that population migration in and out of Wuhan was a primary factor in transmission, which is consistent with much of the literature [18,25,35]. The close relationship between population movements and the prevalence of epidemics has long been established [36]. In the event of an outbreak in a city, the greater the movement of people, the more cases of infection there will be [37]. Transmission dynamics studies have found that the vast majority of cases outside Hubei Province were reported to be from or directly related to Wuhan or Hubei [38]. As the regionally central city and crucial transportation hub in the central region as well as the country, Wuhan has a dense and frequent flow of people. During the disease's spread, most new cases emerged in cities located within Hubei Province (such as Huanggang and Xianning) or in surrounding cities with large migrant populations (such as Xinyang, Chongqing, Changsha, and Hangzhou). These cities, with closer geographical connections and social ties to Wuhan, also shared higher population mobility with Wuhan and were therefore at greater risk of catching and transmitting COVID-19.

Unlike previous studies [3,13,20,24,39], our analysis found that environmental (AQI, humidity, temperature, and PM2.5), demographic (population, density, household, age, ethnicity, gender, education, and urbanicity), socioeconomic (wage, insurance, and unemployment), and other migratory factors (LRB-Bus, Ship, and Air) did not affect epidemic transmission. The weak effect of these factors can likely be explained by the concentrated epidemic pattern in China. Since most new cases were diagnosed in and around Hubei Province—especially in Wuhan, where the epidemic was short-lived—there was not sufficient time for these other factors to influence transmission.

Aside from migration, only the GDP had a significant impact on COVID-19's spread; that is, in the case of fixed migration in Wuhan, regions with higher GDPs were likely to face more serious epidemic transmission. Currently, only a few studies have focused on the role of economic factors in epidemic transmission [30–42]. We considered that a higher GDP was suggestive of greater human activity, closer traffic flows, and socioeconomic connections with other cities, as well as higher population movement (both between and within cities). All the above-mentioned factors can accelerate the interpersonal transmission of COVID-19 and thus increase the risk of diffusion. This conclusion is also consistent with China's disease spread during the early stages. Aside from Hubei Province, cities with higher GDPs (such as Beijing) saw more new diagnoses than other regions. Despite the closer geographical distance, the surrounding cities in Henan, Hunan, Anhui, and other provinces had fewer new cases due to their relatively low social and economic development.

### 4.3. The Transmission Pattern of the Epidemic in China

Using a DSAR model, we further examined the causes of the geographical diffusion and time lag related to COVID-19 to understand better the effect of China's policy on controlling the spread. Compared with other spatiotemporal analysis methods, the DSAR model is more explanatory in mechanism analysis. Methods for detecting spatial heterogeneity have also been used in the analysis of COVID-19, such as multi-scale geographically weighted regression [10,43–48]. However, multi-scale geographically weighted regression currently has no expansion of time–space analysis. Other spatial analysis methods such as 3D bins and emerging hotspots have advantages in prediction but lack transmission mechanisms [49]. We found that the epidemic's transmission in China was affected not only by new case increases in the focal city the day prior but also by those in surrounding cities. Overall, new diagnoses in China were mainly characterized by a combination of inter-city and intra-city transmission.

During the eruption stage of the epidemic, the infection coefficient within cities was 0.76, and between cities it was 0.24. As a result, urban blockades and reductions in population movements were adopted to prevent the rapid spread of the disease. Lockdowns were adopted between cities; at least 18 cities (counties) in Hubei Province took measures to

close the city. Even so, more than five million people left Wuhan before the city was closed, and this massive exodus increased the risk of transmission across the country [50], leading to a massive outbreak. Within cities, residents were restricted from traveling, and public events were canceled to prevent large-scale gathering. Due to the incubation period of COVID-19, however, the number of daily new cases continued to increase in the short term, despite the actions taken by the Chinese government to prevent and control the spread. During the metaphase, intra-city infection was not significant, having a low model R2, while the inter-city infection coefficient was 0.74, which possibly resulted from different statistical methods in different cities.

Moreover, in the anaphase of the epidemic, the infection coefficient was 0.74 within cities and 0 between cities, meaning this phase was dominated by local transmission. Despite the same intra-city infection coefficient as the prophase, the inter-city transmission approached zero and contributed to an overall infection coefficient less than one. The effect of the earlier lockdown in Wuhan became evident in this period; the inter-city transmission gradually stopped as a result of strict measures taken throughout the country to reduce inter-city movement, reducing the movement of persons who possibly had COVID-19. Meanwhile, local transmission had also been effectively controlled [51].

Our study used the DSAR model to explore the spatiotemporal relationship between the number of COVID-19 cases in different cities. We found that in the initial stage of COVID-19, the inter-city transmission coefficient was about one-third that of the intra-city coefficient, and then the inter-city transmission coefficient dropped to zero while the pandemic was controlled. Our research proved the importance of preventing transmission between cities, which has positive policy implications.

## 5. Conclusions

China's strict policies used to control COVID-19's spread, including eliminating inter-city mobility and imposing intra-city travel restrictions (social isolation), were found to be effective ways to reduce the transmission of COVID-19. The findings from this paper confirm that the stoppage of inter-city mobility had the greatest effect on transmission prevention and control. On 23 January, cities in China launched a first-level response to major public health emergencies and took a series of preventive and control measures, including traffic control, migrant management, and the restriction or prohibition of markets and other gathering activities. Some cities with severe outbreaks also adopted lockdown measures, suspending urban public transport, subways, ferries, and other long-distance passenger transport, closing airports and railway stations, among other restrictions, which were considered effective in preventing further transmission.

Although medical researchers have developed therapeutic drugs and vaccines to prevent contracting coronavirus, reasonable social isolation measures are still largely effective in controlling the spread of COVID-19 [52]. The lockdown and the "reduction of population mobility" policies are both of great significance in controlling epidemic transmission. Importantly, policies should also vary from phase to phase in response to the different spatial and temporal characteristics of disease spread. The global epidemic situation has entered a normalization period of prevention and control but needs all countries and cities' joint efforts to suppress the spread of COVID-19 completely.

**Author Contributions:** Conceptualization, Hanchen Yu and Hengyu Gu; methodology, Hanchen Yu; software, Hanchen Yu; validation; formal analysis, Hanchen Yu; writing—original draft preparation, Jingwei Li; writing—review and editing, Sarah Bardin; visualization, Hanchen Yu; supervision, Chenjing Fan. All authors have read and agreed to the published version of the manuscript.

**Funding:** This research was funded by the Major Program of the National Social Science Foundation of China, grant number 17ZDA055.

**Institutional Review Board Statement:** Not applicable.

**Informed Consent Statement:** Not applicable.

**Data Availability Statement:** The data that support the findings of this study are available from the corresponding author upon reasonable request.

**Conflicts of Interest:** The authors declare no conflict of interest.

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
