# Peer review of "Spatiotemporal Dynamic of COVID-19 Diffusion in China: A Dynamic Spatial Autoregressive Model Analysis"

_ijgi, doi:10.3390/ijgi10080510_

Round 1

Reviewer 1 Report

The authors looked at spatial and temporal patterns of COVID spread in China. They found that social isolation is the best way to reduce transmission. 

The findings of this paper do not seem to be novel. The authors need to do a better job of pointing out the novelty of their study. They point out that GDP had an effect on the number of cases. But nothing is made of that later in the discussion and conclusions. Instead, the authors need to make a clear point on how their results are novel from other groups, or whether their methods are novel, even though the conclusions are not very different from the rest. 

In other words, please make it clear in the conclusions how your study is novel compared to the literature. 

Reviewer 2 Report

The article "Spatio-Temporal Dynamic of COVID-19 Diffusion in China: A 2 Dynamic Spatial Autoregressive Model Analysis" describes the mechanism of COVID-19 transmission in different periods within and between cities, in the case study of China.

The paper is well structured, it follows an academic structure. It describes a hot topic. The workflow and methodology is not unique but still adeqaute for publishing. 

Except seven maps, I do not see any GIS implementation. Which softtware, which algorithsm, data source etc. was used?

Figure 1 - if image illustrates daily average, then indication of dates (from-to) is missing. Honestly, I do not trust this image. According to Fig 1 it seems to me that within half of China there was only few (less then 4) people per a day indicated with covid. Is it really true? In general, accordign your maps it seems to be no covid except a few cities in China (e.g. Fig2f - only 12 new cases is a total maximum for China? Really?).

I definitelly not agree with a statements like "While China was able to control the spread of 14 COVID-19 within the country swiftly, many other countries continue to struggle to contain the out-15 break over a year later." (line 14-15) or "Nonetheless, in countries like the United States, the United Kingdom, and Russia, the epidemic has not been well-controlled; the number of newly confirmed cases has not yet decreased significantly even after the implementation of social isolation measures" (391-393). This is a non-objective, unsubstantiated, moreover according to my opinion politically-oriented . Therefore statement this statements are unacceptable for academic paper.

Reviewer 3 Report

The research is focused on an interesting topic. Nowadays, the analysis of the not only temporal but also spatial evolution of the pandemic is a relevant research field. The authors propose a balanced research between statistical and cartographic approach and from the point of view of basic and applied research, to contribute to the taking decision process.

Nevertheless, it is important to highlight that the research implies a high and qualified use of GIS, nevertheless the importance of GIS or geotechnologies is hidden in the manuscript. This is a minor detail, but it is related to other unusual characteristic of the paper: it is centred on a spatial approach, but the territory seems in second position regarding to the interpretation of results. It is an important defect (or lack of interdisciplinary perspective), the authors should improve. If the study produces a relevant spatial pattern, it would be better linked to the context where it happens.

Despite this initial vision of the manuscript, I will explain below several contents that the authors should correct, explain, or improve.

An important limitation I find is the too short period time the study analyses. Other studies about COVID-19 temporal and spatial patterns consider several months, but in this case the study is based in less than two months. Really, is it possible to stablish three stages? What about future evolution after the declination stage? One year later the pandemic continues. How can contribute the study to control measures if the conclusions are based on the evolution one year ago? I suggest the authors a consideration about this. At least they should mention it in the discussion part.

Other important issue in this research is related to the methodology. An interesting background about other methodological approaches about space-time analysis is omitted. The authors use a linear regression to model spatio-temporal dynamics. Nevertheless, other methodologies have been applied to similar goals related to Covid (as 3D bins and emerging analysis, among others). At least, they should consider it in discussion section to compare approaches. On the other hand, many studies about diseases distribution hold the idea the non-stationarity from geographic perspective. Consequently, many studies include the weighted linear regression. The authors should mention it, at leas in discussion and value the benefits of their method in comparison with other extended methodologies in the spatial and temporal perspective of the pandemic.

Conclusion

It is strange that the authors refer to other countries in relation to the pandemic control. It is my impression that it is not related to the rest of the study, therefore conclusion should present a better alignment with the specific topic of the manuscript of own results.  

Maps should be improved:

  • Improve quality because figures seem pixelated.
  • Legends should include the spatial delimitations of the map (administrative boundaries level). The boundary of Hainan is wider?
  • It should be helpful the labels of the location of main cities for readers from other countries.
  • Legend intervals: First interval goes from 0.00 to “n”. Regarding to minimum value I suppose that it should be >0.00. Is null value in wite colour interpreted as not cases? Then, the first interval should be always from some cases to…

Regarding to the evolution maps (Figure 2), it is impossible to compare spatial patterns because the legend intervals change from one map to other. The authors should define only one legend and analyse the changes by day with the same conditions (intervals). I know that some days will not include units in a part of the intervals, but if someone want to compare spatial distribution along time, legend must be invariable.

Round 2

Reviewer 2 Report

Authors incorporate changes based on the review. Authors discussed the reasons in the cover letter succesfully. Therefore I reccomend to accept the paper for publishing in IJGI journal. Before publishin, please focus on images, in current version images are in very low quality

Reviewer 3 Report

I consider the authors have improved the initial version of the manuscript. Methodology background has a wider approach in present version. They improved the cartography. In conclusion, the authors had in consideration each suggestion and comment and solved it correctly.

Only one minimum detail: I observed that in new text they wrote they write covid-19 in different format that in the previous text (capital letters). So, they should adapt it and write in a homogeneous format.